# DATASET CONDENSATION WITH GRADIENT MATCHING

**Bo Zhao, Konda Reddy Mopuri, Hakan Bilen**
School of Informatics, The University of Edinburgh
`{bo.zhao, kmopuri, hbilen}@ed.ac.uk`

## ABSTRACT

As the state-of-the-art machine learning methods in many fields rely on larger datasets, storing datasets and training models on them become significantly more expensive. This paper proposes a training set synthesis technique for *data-efficient* learning, called *Dataset Condensation*, that learns to condense large dataset into a small set of informative synthetic samples for training deep neural networks from scratch. We formulate this goal as a gradient matching problem between the gradients of deep neural network weights that are trained on the original and our synthetic data. We rigorously evaluate its performance in several computer vision benchmarks and demonstrate that it significantly outperforms the state-of-the-art methods[1]. Finally we explore the use of our method in continual learning and neural architecture search and report promising gains when limited memory and computations are available.

## 1 INTRODUCTION

Large-scale datasets, comprising millions of samples, are becoming the norm to obtain state-of-the-art machine learning models in multiple fields including computer vision, natural language processing and speech recognition. At such scales, even storing and preprocessing the data becomes burdensome, and training machine learning models on them demands for specialized equipment and infrastructure. An effective way to deal with large data is data selection – identifying the most representative training samples – that aims at improving *data efficiency* of machine learning techniques. While classical data selection methods, also known as coreset construction (Agarwal et al., 2004; Har-Peled & Mazumdar, 2004; Feldman et al., 2013), focus on clustering problems, recent work can be found in continual learning (Rebuffi et al., 2017; Toneva et al., 2019; Castro et al., 2018; Aljundi et al., 2019) and active learning (Sener & Savarese, 2018) where there is typically a fixed budget in storing and labeling training samples respectively. These methods commonly first define a criterion for representativeness (*e.g.* in terms of compactness (Rebuffi et al., 2017; Castro et al., 2018), diversity (Sener & Savarese, 2018; Aljundi et al., 2019), forgetfulness (Toneva et al., 2019)), then select the representative samples based on the criterion, finally use the selected small set to train their model for a downstream task.

Unfortunately, these methods have two shortcomings: they typically rely on i) heuristics (*e.g.* picking cluster centers) that does not guarantee any optimal solution for the downstream task (*e.g.* image classification), ii) presence of representative samples, which is neither guaranteed. A recent method, Dataset Distillation (DD) (Wang et al., 2018) goes beyond these limitations by *learning* a small set of informative images from large training data. In particular, the authors model the network parameters as a function of the synthetic training data and learn them by minimizing the training loss over the original training data w.r.t. synthetic data. Unlike in the coreset methods, the synthesized data are directly optimized for the downstream task and thus the success of the method does not rely on the presence of representative samples.

Inspired from DD (Wang et al., 2018), we focus on learning to *synthesize informative samples* that are optimized to train neural networks for downstream tasks and not limited to individual samples in original dataset. Like DD, our goal is to obtain the highest generalization performance with a model trained on a small set of synthetic images, ideally comparable performance to that of a model trained on the original images (see Figure 1(a)). In particular, we investigate the following

---

[1]The implementation is available at `https://github.com/VICO-UoE/DatasetCondensation`.

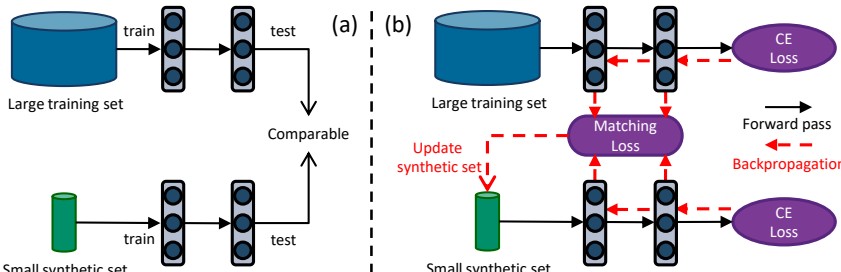

Figure 1: Dataset Condensation (left) aims to generate a small set of synthetic images that can match the performance of a network trained on a large image dataset. Our method (right) realizes this goal by learning a synthetic set such that a deep network trained on it and the large set produces similar gradients w.r.t. its weights. The synthetic data can later be used to train a network from scratch in a small fraction of the original computational load. CE denotes Cross-Entropy.

questions. Is it possible to i) compress a large image classification dataset into a small synthetic set, ii) train an image classification model on the synthetic set that can be further used to classify real images, iii) learn a single set of synthetic images that can be used to train different neural network architectures? To this end, we propose a *Dataset Condensation* method to learn a small set of "condensed" synthetic samples such that a deep neural network trained on them obtains not only similar performance but also a close solution to a network trained on the large training data in the network parameter space. We formulate this goal as a minimization problem between two sets of gradients of the network parameters that are computed for a training loss over a large fixed training set and a learnable condensed set (see Figure 1(b)). We show that our method enables effective learning of synthetic images and neural networks trained on them, outperforms (Wang et al., 2018) and coreset methods with a wide margin in multiple computer vision benchmarks. In addition, learning a compact set of synthetic samples also benefits other learning problems when there is a fixed budget on training images. We show that our method outperforms popular data selection methods by providing more informative training samples in continual learning. Finally, we explore a promising use case of our method in neural architecture search, and show that – once our condensed images are learned – they can be used to train numerous network architectures extremely efficiently.

Our method is related to knowledge distillation (KD) techniques (Hinton et al., 2015; Buciluă et al., 2006; Ba & Caruana, 2014; Romero et al., 2014) that transfer the knowledge in an ensemble of models to a single one. Unlike KD, we distill knowledge of a large training set into a small synthetic set. Our method is also related to Generative Adversarial Networks (Goodfellow et al., 2014a; Mirza & Osindero, 2014; Radford et al., 2015) and Variational AutoEncoders (Kingma & Welling, 2013) that synthesize high-fidelity samples by capturing the data distribution. In contrast, our goal is to generate informative samples for training deep neural networks rather than to produce "real-looking" samples. Finally our method is related to the methods that produce image patches by projecting the feature activations back to the input pixel space (Zeiler & Fergus, 2014), reconstruct the input image by matching the feature activations (Mahendran & Vedaldi, 2015), recover private training images for given training gradients (Zhu et al., 2019; Zhao et al., 2020), synthesize features from semantic embeddings for zero-shot learning (Sariyildiz & Cinbis, 2019). Our goal is however to synthesize a set of condensed training images not to recover the original or missing training images.

In the remainder of this paper, we first review the problem of dataset condensation and introduce our method in section 2, present and analyze our results in several image recognition benchmarks in section 3.1, showcase applications in continual learning and network architecture search in section 3.2, and conclude the paper with remarks for future directions in section 4.

## 2 METHOD

### 2.1 DATASET CONDENSATION

Suppose we are given a large dataset consisting of $|\mathcal{T}|$ pairs of a training image and its class label $\mathcal{T} = \{(\boldsymbol{x}_i, y_i)\}\big|_{i=1}^{|\mathcal{T}|}$ where $\boldsymbol{x} \in \mathcal{X} \subset \mathbb{R}^d$, $y \in \{0, \ldots, C-1\}$, $\mathcal{X}$ is a d-dimensional input space and $C$ is the number of classes. We wish to learn a differentiable function $\phi$ (*i.e.* deep neural network)

with parameters $\boldsymbol{\theta}$ that correctly predicts labels of previously unseen images, *i.e.* $y = \phi_{\boldsymbol{\theta}}(\boldsymbol{x})$. One can learn the parameters of this function by minimizing an empirical loss term over the training set:

$$\boldsymbol{\theta}^{\mathcal{T}} = \underset{\boldsymbol{\theta}}{\arg\min}\, \mathcal{L}^{\mathcal{T}}(\boldsymbol{\theta}) \tag{1}$$

where $\mathcal{L}^{\mathcal{T}}(\boldsymbol{\theta}) = \frac{1}{|\mathcal{T}|} \sum_{(\boldsymbol{x},y) \in \mathcal{T}} \ell(\phi_{\boldsymbol{\theta}}(\boldsymbol{x}), y)$ , $\ell(\cdot, \cdot)$ is a task specific loss (*i.e.* cross-entropy) and $\boldsymbol{\theta}^{\mathcal{T}}$ is the minimizer of $\mathcal{L}^{\mathcal{T}}$. The generalization performance of the obtained model $\phi_{\boldsymbol{\theta}^{\mathcal{T}}}$ can be written as $\mathbb{E}_{\boldsymbol{x} \sim P_{\mathcal{D}}}[\ell(\phi_{\boldsymbol{\theta}^{\mathcal{T}}}(\boldsymbol{x}), y)]$ where $P_{\mathcal{D}}$ is the data distribution. Our goal is to generate a small set of condensed synthetic samples with their labels, $\mathcal{S} = \{(\boldsymbol{s}_i, y_i)\}|_{i=1}^{|\mathcal{S}|}$ where $\boldsymbol{s} \in \mathbb{R}^d$ and $y \in \mathcal{Y}$, $|\mathcal{S}| \ll |\mathcal{T}|$. Similar to eq. (1), once the condensed set is learned, one can train $\phi$ on them as follows

$$\boldsymbol{\theta}^{\mathcal{S}} = \underset{\boldsymbol{\theta}}{\arg\min}\, \mathcal{L}^{\mathcal{S}}(\boldsymbol{\theta}) \tag{2}$$

where $\mathcal{L}^{\mathcal{S}}(\boldsymbol{\theta}) = \frac{1}{|\mathcal{S}|} \sum_{(\boldsymbol{s},y) \in \mathcal{S}} \ell(\phi_{\boldsymbol{\theta}}(\boldsymbol{s}), y)$ and $\boldsymbol{\theta}^{\mathcal{S}}$ is the minimizer of $\mathcal{L}^{\mathcal{S}}$. As the synthetic set $\mathcal{S}$ is significantly smaller (2-3 orders of magnitude), we expect the optimization in eq. (2) to be significantly faster than that in eq. (1). We also wish the generalization performance of $\phi_{\boldsymbol{\theta}^{\mathcal{S}}}$ to be close to $\phi_{\boldsymbol{\theta}^{\mathcal{T}}}$, *i.e.* $\mathbb{E}_{\boldsymbol{x} \sim P_{\mathcal{D}}}[\ell(\phi_{\boldsymbol{\theta}^{\mathcal{T}}}(\boldsymbol{x}), y)] \simeq \mathbb{E}_{\boldsymbol{x} \sim P_{\mathcal{D}}}[\ell(\phi_{\boldsymbol{\theta}^{\mathcal{S}}}(\boldsymbol{x}), y)]$ over the real data distribution $P_{\mathcal{D}}$.

**Discussion.** The goal of obtaining comparable generalization performance by training on the condensed data can be formulated in different ways. One approach, which is proposed in (Wang et al., 2018) and extended in (Sucholutsky & Schonlau, 2019; Bohdal et al., 2020; Such et al., 2020), is to pose the parameters $\boldsymbol{\theta}^{\mathcal{S}}$ as a function of the synthetic data $\mathcal{S}$:

$$\mathcal{S}^* = \underset{\mathcal{S}}{\arg\min}\, \mathcal{L}^{\mathcal{T}}(\boldsymbol{\theta}^{\mathcal{S}}(\mathcal{S})) \qquad \text{subject to} \qquad \boldsymbol{\theta}^{\mathcal{S}}(\mathcal{S}) = \underset{\boldsymbol{\theta}}{\arg\min}\, \mathcal{L}^{\mathcal{S}}(\boldsymbol{\theta}). \tag{3}$$

The method aims to find the optimum set of synthetic images $\mathcal{S}^*$ such that the model $\phi_{\boldsymbol{\theta}^{\mathcal{S}}}$ trained on them minimizes the training loss over the original data. Optimizing eq. (3) involves a nested loop optimization and solving the inner loop for $\boldsymbol{\theta}^{\mathcal{S}}(\mathcal{S})$ at each iteration to recover the gradients for $\mathcal{S}$ which requires a computationally expensive procedure – unrolling the recursive computation graph for $\mathcal{S}$ over multiple optimization steps for $\boldsymbol{\theta}$ (see (Samuel & Tappen, 2009; Domke, 2012)). Hence, it does not scale to large models and/or accurate inner-loop optimizers with many steps. Next we propose an alternative formulation for dataset condensation.

## 2.2 DATASET CONDENSATION WITH PARAMETER MATCHING

Here we aim to learn $\mathcal{S}$ such that the model $\phi_{\boldsymbol{\theta}^{\mathcal{S}}}$ trained on them achieves not only comparable generalization performance to $\phi_{\boldsymbol{\theta}^{\mathcal{T}}}$ but also converges to a similar solution in the parameter space (*i.e.* $\boldsymbol{\theta}^{\mathcal{S}} \approx \boldsymbol{\theta}^{\mathcal{T}}$). Let $\phi_{\boldsymbol{\theta}}$ be a locally smooth function[2], similar weights ($\boldsymbol{\theta}^{\mathcal{S}} \approx \boldsymbol{\theta}^{\mathcal{T}}$) imply similar mappings in a local neighborhood and thus generalization performance, *i.e.* $\mathbb{E}_{\boldsymbol{x} \sim P_{\mathcal{D}}}[\ell(\phi_{\boldsymbol{\theta}^{\mathcal{T}}}(\boldsymbol{x}), y)] \simeq \mathbb{E}_{\boldsymbol{x} \sim P_{\mathcal{D}}}[\ell(\phi_{\boldsymbol{\theta}^{\mathcal{S}}}(\boldsymbol{x}), y)]$. Now we can formulate this goal as

$$\underset{\mathcal{S}}{\min}\, D(\boldsymbol{\theta}^{\mathcal{S}}, \boldsymbol{\theta}^{\mathcal{T}}) \quad \text{subject to} \quad \boldsymbol{\theta}^{\mathcal{S}}(\mathcal{S}) = \underset{\boldsymbol{\theta}}{\arg\min}\, \mathcal{L}^{\mathcal{S}}(\boldsymbol{\theta}) \tag{4}$$

where $\boldsymbol{\theta}^{\mathcal{T}} = \arg\min_{\boldsymbol{\theta}} \mathcal{L}^{\mathcal{T}}(\boldsymbol{\theta})$ and $D(\cdot, \cdot)$ is a distance function. In a deep neural network, $\boldsymbol{\theta}^{\mathcal{T}}$ typically depends on its initial values $\boldsymbol{\theta}_0$. However, the optimization in eq. (4) aims to obtain an optimum set of synthetic images only for one model $\phi_{\boldsymbol{\theta}^{\mathcal{T}}}$ with the initialization $\boldsymbol{\theta}_0$, while our actual goal is to generate samples that can work with a distribution of random initializations $P_{\boldsymbol{\theta}_0}$. Thus we modify eq. (4) as follows:

$$\underset{\mathcal{S}}{\min}\, \mathrm{E}_{\boldsymbol{\theta}_0 \sim P_{\boldsymbol{\theta}_0}}[D(\boldsymbol{\theta}^{\mathcal{S}}(\boldsymbol{\theta}_0), \boldsymbol{\theta}^{\mathcal{T}}(\boldsymbol{\theta}_0))] \quad \text{subject to} \quad \boldsymbol{\theta}^{\mathcal{S}}(\mathcal{S}) = \underset{\boldsymbol{\theta}}{\arg\min}\, \mathcal{L}^{\mathcal{S}}(\boldsymbol{\theta}(\boldsymbol{\theta}_0)) \tag{5}$$

where $\boldsymbol{\theta}^{\mathcal{T}} = \arg\min_{\boldsymbol{\theta}} \mathcal{L}^{\mathcal{T}}(\boldsymbol{\theta}(\boldsymbol{\theta}_0))$. For brevity, we use only $\boldsymbol{\theta}^{\mathcal{S}}$ and $\boldsymbol{\theta}^{\mathcal{T}}$ to indicate $\boldsymbol{\theta}^{\mathcal{S}}(\boldsymbol{\theta}_0)$ and $\boldsymbol{\theta}^{\mathcal{T}}(\boldsymbol{\theta}_0)$ respectively in the next sections. The standard approach to solving eq. (5) employs implicit differentiation (see (Domke, 2012) for details), which involves solving an inner loop optimization for $\boldsymbol{\theta}^{\mathcal{S}}$. As the inner loop optimization $\boldsymbol{\theta}^{\mathcal{S}}(\mathcal{S}) = \arg\min_{\boldsymbol{\theta}} \mathcal{L}^{\mathcal{S}}(\boldsymbol{\theta})$ can be computationally expensive in

---

[2]Local smoothness is frequently used to obtain explicit first-order local approximations in deep networks (*e.g.* see (Rifai et al., 2012; Goodfellow et al., 2014b; Koh & Liang, 2017)).

case of large-scale models, one can adopt the back-optimization approach in (Domke, 2012) which re-defines $\boldsymbol{\theta}^{\mathcal{S}}$ as the output of an incomplete optimization:

$$\boldsymbol{\theta}^{\mathcal{S}}(\mathcal{S}) = \text{opt-alg}_{\boldsymbol{\theta}}(\mathcal{L}^{\mathcal{S}}(\boldsymbol{\theta}), \varsigma) \tag{6}$$

where $\text{opt-alg}$ is a specific optimization procedure with a fixed number of steps ($\varsigma$).

In practice, $\boldsymbol{\theta}^{\mathcal{T}}$ for different initializations can be trained first in an offline stage and then used as the target parameter vector in eq. (5). However, there are two potential issues by learning to regress $\boldsymbol{\theta}^{\mathcal{T}}$ as the target vector. First the distance between $\boldsymbol{\theta}^{\mathcal{T}}$ and intermediate values of $\boldsymbol{\theta}^{\mathcal{S}}$ can be too big in the parameter space with multiple local minima traps along the path and thus it can be too challenging to reach. Second $\text{opt-alg}$ involves a limited number of optimization steps as a trade-off between speed and accuracy which may not be sufficient to take enough steps for reaching the optimal solution. These problems are similar to those of (Wang et al., 2018), as they both involve parameterizing $\boldsymbol{\theta}^{\mathcal{S}}$ with $\mathcal{S}$ and $\boldsymbol{\theta}_0$.

## 2.3 DATASET CONDENSATION WITH CURRICULUM GRADIENT MATCHING

Here we propose a curriculum based approach to address the above mentioned challenges. The key idea is that we wish $\boldsymbol{\theta}^{\mathcal{S}}$ to be close to not only the final $\boldsymbol{\theta}^{\mathcal{T}}$ but also to follow a similar path to $\boldsymbol{\theta}^{\mathcal{T}}$ throughout the optimization. While this can restrict the optimization dynamics for $\boldsymbol{\theta}$, we argue that it also enables a more guided optimization and effective use of the incomplete optimizer. We can now decompose eq. (5) into multiple subproblems:

$$\min_{\mathcal{S}} \mathrm{E}_{\boldsymbol{\theta}_0 \sim P_{\boldsymbol{\theta}_0}} \big[ \sum_{t=0}^{T-1} D(\boldsymbol{\theta}_t^{\mathcal{S}}, \boldsymbol{\theta}_t^{\mathcal{T}}) \big] \quad \text{subject to}$$
$$\boldsymbol{\theta}_{t+1}^{\mathcal{S}}(\mathcal{S}) = \text{opt-alg}_{\boldsymbol{\theta}}(\mathcal{L}^{\mathcal{S}}(\boldsymbol{\theta}_t^{\mathcal{S}}), \varsigma^{\mathcal{S}}) \quad \text{and} \quad \boldsymbol{\theta}_{t+1}^{\mathcal{T}} = \text{opt-alg}_{\boldsymbol{\theta}}(\mathcal{L}^{\mathcal{T}}(\boldsymbol{\theta}_t^{\mathcal{T}}), \varsigma^{\mathcal{T}}) \tag{7}$$

where $T$ is the number of iterations, $\varsigma^{\mathcal{S}}$ and $\varsigma^{\mathcal{T}}$ are the numbers of optimization steps for $\boldsymbol{\theta}^{\mathcal{S}}$ and $\boldsymbol{\theta}^{\mathcal{T}}$ respectively. In words, we wish to generate a set of condensed samples $\mathcal{S}$ such that the network parameters trained on them ($\boldsymbol{\theta}_t^{\mathcal{S}}$) are similar to the ones trained on the original training set ($\boldsymbol{\theta}_t^{\mathcal{T}}$) at each iteration $t$. In our preliminary experiments, we observe that $\boldsymbol{\theta}_{t+1}^{\mathcal{S}}$, which is parameterized with $\mathcal{S}$, can successfully track $\boldsymbol{\theta}_{t+1}^{\mathcal{T}}$ by updating $\mathcal{S}$ and minimizing $D(\boldsymbol{\theta}_t^{\mathcal{S}}, \boldsymbol{\theta}_t^{\mathcal{T}})$ close to zero.

In the case of one step gradient descent optimization for $\text{opt-alg}$, the update rule is:

$$\boldsymbol{\theta}_{t+1}^{\mathcal{S}} \leftarrow \boldsymbol{\theta}_t^{\mathcal{S}} - \eta_{\boldsymbol{\theta}} \nabla_{\boldsymbol{\theta}} \mathcal{L}^{\mathcal{S}}(\boldsymbol{\theta}_t^{\mathcal{S}}) \quad \text{and} \quad \boldsymbol{\theta}_{t+1}^{\mathcal{T}} \leftarrow \boldsymbol{\theta}_t^{\mathcal{T}} - \eta_{\boldsymbol{\theta}} \nabla_{\boldsymbol{\theta}} \mathcal{L}^{\mathcal{T}}(\boldsymbol{\theta}_t^{\mathcal{T}}), \tag{8}$$

where $\eta_{\boldsymbol{\theta}}$ is the learning rate. Based on our observation ($D(\boldsymbol{\theta}_t^{\mathcal{S}}, \boldsymbol{\theta}_t^{\mathcal{T}}) \approx 0$), we simplify the formulation in eq. (7) by replacing $\boldsymbol{\theta}_t^{\mathcal{T}}$ with $\boldsymbol{\theta}_t^{\mathcal{S}}$ and use $\boldsymbol{\theta}$ to denote $\boldsymbol{\theta}^{\mathcal{S}}$ in the rest of the paper:

$$\min_{\mathcal{S}} \mathrm{E}_{\boldsymbol{\theta}_0 \sim P_{\boldsymbol{\theta}_0}} \big[ \sum_{t=0}^{T-1} D(\nabla_{\boldsymbol{\theta}} \mathcal{L}^{\mathcal{S}}(\boldsymbol{\theta}_t), \nabla_{\boldsymbol{\theta}} \mathcal{L}^{\mathcal{T}}(\boldsymbol{\theta}_t)) \big]. \tag{9}$$

We now have a single deep network with parameters $\boldsymbol{\theta}$ trained on the synthetic set $\mathcal{S}$ which is optimized such that the distance between the gradients for the loss over the training samples $\mathcal{L}^{\mathcal{T}}$ w.r.t. $\boldsymbol{\theta}$ and the gradients for the loss over the condensed samples $\mathcal{L}^{\mathcal{S}}$ w.r.t. $\boldsymbol{\theta}$ is minimized. In words, our goal reduces to matching the gradients for the real and synthetic training loss w.r.t. $\boldsymbol{\theta}$ via updating the condensed samples. This approximation has the key advantage over (Wang et al., 2018) and eq. (5) that it does not require the expensive unrolling of the recursive computation graph over the previous parameters $\{\boldsymbol{\theta}_0, \ldots, \boldsymbol{\theta}_{t-1}\}$. The important consequence is that the optimization is significantly faster, memory efficient and thus scales up to the state-of-the-art deep neural networks (*e.g.* ResNet (He et al., 2016)).

**Discussion.** The synthetic data contains not only samples but also their labels $(\boldsymbol{s}, y)$ that can be jointly learned by optimizing eq. (9) in theory. However, their joint optimization is challenging, as the content of the samples depend on their label and vice-versa. Thus in our experiments we learn to synthesize images for fixed labels, *e.g.* one synthetic image per class.

**Algorithm.**    We depict the optimization details in Alg. 1. At the outer level, it contains a loop over random weight initializations, as we want to obtain condensed images that can later be used to train previously unseen models. Once $\boldsymbol{\theta}$ is randomly initialized, we use $\phi_{\boldsymbol{\theta}}$ to first compute the loss over both the training samples ($\mathcal{L}^{\mathcal{T}}$), synthetic samples ($\mathcal{L}^{\mathcal{S}}$) and their gradients w.r.t. $\boldsymbol{\theta}$, then optimize the synthetic samples $\mathcal{S}$ to match these gradients $\nabla_{\boldsymbol{\theta}}\mathcal{L}^{\mathcal{S}}$ to $\nabla_{\boldsymbol{\theta}}\mathcal{L}^{\mathcal{T}}$ by applying $\varsigma_{\mathcal{S}}$ gradient descent steps with learning rate $\eta_{\mathcal{S}}$. We use the stochastic gradient descent optimization for both opt-alg$_{\boldsymbol{\theta}}$ and opt-alg$_{\mathcal{S}}$. Next we train $\boldsymbol{\theta}$ on the updated synthetic images by minimizing the loss $\mathcal{L}^{\mathcal{S}}$ with learning rate $\eta_{\boldsymbol{\theta}}$ for $\varsigma_{\boldsymbol{\theta}}$ steps. Note that we sample each real and synthetic batch pair from $\mathcal{T}$ and $\mathcal{S}$ containing samples from a single class and the synthetic data for each class are separately (or parallelly) updated at each iteration ($t$) for the following reasons: i) this reduces memory use at train time, ii) imitating the mean gradients w.r.t. the data from single class is easier compared to those of multiple classes. This does not bring any extra computational cost.

---

**Algorithm 1:** Dataset condensation with gradient matching

---

**Input:** Training set $\mathcal{T}$

1 **Required**: Randomly initialized set of synthetic samples $\mathcal{S}$ for $C$ classes, probability distribution over randomly initialized weights $P_{\boldsymbol{\theta}_0}$, deep neural network $\phi_{\boldsymbol{\theta}}$, number of outer-loop steps $K$, number of inner-loop steps $T$, number of steps for updating weights $\varsigma_{\boldsymbol{\theta}}$ and synthetic samples $\varsigma_{\mathcal{S}}$ in each inner-loop step respectively, learning rates for updating weights $\eta_{\boldsymbol{\theta}}$ and synthetic samples $\eta_{\mathcal{S}}$.

2 **for** $k = 0, \cdots, K - 1$ **do**

3     Initialize $\boldsymbol{\theta}_0 \sim P_{\boldsymbol{\theta}_0}$

4     **for** $t = 0, \cdots, T - 1$ **do**

5         **for** $c = 0, \cdots, C - 1$ **do**

6             Sample a minibatch pair $B_c^{\mathcal{T}} \sim \mathcal{T}$ and $B_c^{\mathcal{S}} \sim \mathcal{S}$     ▷ $B_c^{\mathcal{T}}$ and $B_c^{\mathcal{S}}$ are of the same class $c$.

7             Compute $\mathcal{L}_c^{\mathcal{T}} = \frac{1}{|B_c^{\mathcal{T}}|} \sum_{(\boldsymbol{x},y) \in B_c^{\mathcal{T}}} \ell(\phi_{\boldsymbol{\theta}_t}(\boldsymbol{x}), y)$ and $\mathcal{L}_c^{\mathcal{S}} = \frac{1}{|B_c^{\mathcal{S}}|} \sum_{(\boldsymbol{s},y) \in B_c^{\mathcal{S}}} \ell(\phi_{\boldsymbol{\theta}_t}(\boldsymbol{s}), y)$

8             Update $\mathcal{S}_c \leftarrow$ opt-alg$_{\mathcal{S}}(D(\nabla_{\boldsymbol{\theta}}\mathcal{L}_c^{\mathcal{S}}(\boldsymbol{\theta}_t), \nabla_{\boldsymbol{\theta}}\mathcal{L}_c^{\mathcal{T}}(\boldsymbol{\theta}_t)), \varsigma_{\mathcal{S}}, \eta_{\mathcal{S}})$

9         Update $\boldsymbol{\theta}_{t+1} \leftarrow$ opt-alg$_{\boldsymbol{\theta}}(\mathcal{L}^{\mathcal{S}}(\boldsymbol{\theta}_t), \varsigma_{\boldsymbol{\theta}}, \eta_{\boldsymbol{\theta}})$     ▷ Use the whole $\mathcal{S}$

**Output:** $\mathcal{S}$

---

**Gradient matching loss.**    The matching loss $D(\cdot, \cdot)$ in eq. (9) measures the distance between the gradients for $\mathcal{L}^{\mathcal{S}}$ and $\mathcal{L}^{\mathcal{T}}$ w.r.t. $\boldsymbol{\theta}$. When $\phi_{\boldsymbol{\theta}}$ is a multi-layered neural network, the gradients correspond to a set of learnable 2D (out$\times$in) and 4D (out$\times$in$\times$h$\times$w) weights for each fully connected (FC) and convolutional layer resp where out, in, h, w are number of output and input channels, kernel height and width resp. The matching loss can be decomposed into a sum of layerwise losses as $D(\nabla_{\boldsymbol{\theta}}\mathcal{L}^{\mathcal{S}}, \nabla_{\boldsymbol{\theta}}\mathcal{L}^{\mathcal{T}}) = \sum_{l=1}^{L} d(\nabla_{\boldsymbol{\theta}^{(l)}}\mathcal{L}^{\mathcal{S}}, \nabla_{\boldsymbol{\theta}^{(l)}}\mathcal{L}^{\mathcal{T}})$ where $l$ is the layer index, $L$ is the number of layers with weights and

$$d(\mathbf{A}, \mathbf{B}) = \sum_{i=1}^{\text{out}} \left( 1 - \frac{\mathbf{A}_{\mathbf{i}\cdot} \cdot \mathbf{B}_{\mathbf{i}\cdot}}{\|\mathbf{A}_{\mathbf{i}\cdot}\|\|\mathbf{B}_{\mathbf{i}\cdot}\|} \right) \tag{10}$$

where $\mathbf{A}_{i\cdot}$ and $\mathbf{B}_{i\cdot}$ are flattened vectors of gradients corresponding to each output node $i$, which is in dimensional for FC weights and in$\times$h$\times$w dimensional for convolutional weights. In contrast to (Lopez-Paz et al., 2017; Aljundi et al., 2019; Zhu et al., 2019) that ignore the layer-wise structure by flattening tensors over all layers to one vector and then computing the distance between two vectors, we group them for each output node. We found that this is a better distance for gradient matching (see the supplementary) and enables using a single learning rate across all layers.

## 3 EXPERIMENTS

### 3.1 DATASET CONDENSATION

First we evaluate classification performance with the condensed images on four standard benchmark datasets: digit recognition on MNIST (LeCun et al., 1998), SVHN (Netzer et al., 2011) and object classification on FashionMNIST (Xiao et al., 2017), CIFAR10 (Krizhevsky et al., 2009). We test our method using six standard deep network architectures: MLP, ConvNet (Gidaris & Komodakis, 2018), LeNet (LeCun et al., 1998), AlexNet (Krizhevsky et al., 2012), VGG-11 (Simonyan & Zisserman, 2014) and ResNet-18 (He et al., 2016). MLP is a multilayer perceptron with two nonlinear hidden layers, each has 128 units. ConvNet is a commonly used modular architecture in few-shot

| | Img/Cls | Ratio % | Random | Coreset Selection Herding | K-Center | Forgetting | Ours | Whole Dataset |
|---|---|---|---|---|---|---|---|---|
| MNIST | 1 | 0.017 | 64.9±3.5 | 89.2±1.6 | 89.3±1.5 | 35.5±5.6 | **91.7±0.5** | |
| | 10 | 0.17 | 95.1±0.9 | 93.7±0.3 | 84.4±1.7 | 68.1±3.3 | **97.4±0.2** | 99.6±0.0 |
| | 50 | 0.83 | 97.9±0.2 | 94.9±0.2 | 97.4±0.3 | 88.2±1.2 | **98.8±0.2** | |
| FashionMNIST | 1 | 0.017 | 51.4±3.8 | 67.0±1.9 | 66.9±1.8 | 42.0±5.5 | **70.5±0.6** | |
| | 10 | 0.17 | 73.8±0.7 | 71.1±0.7 | 54.7±1.5 | 53.9±2.0 | **82.3±0.4** | 93.5±0.1 |
| | 50 | 0.83 | 82.5±0.7 | 71.9±0.8 | 68.3±0.8 | 55.0±1.1 | **83.6±0.4** | |
| SVHN | 1 | 0.014 | 14.6±1.6 | 20.9±1.3 | 21.0±1.5 | 12.1±1.7 | **31.2±1.4** | |
| | 10 | 0.14 | 35.1±4.1 | 50.5±3.3 | 14.0±1.3 | 16.8±1.2 | **76.1±0.6** | 95.4±0.1 |
| | 50 | 0.7 | 70.9±0.9 | 72.6±0.8 | 20.1±1.4 | 27.2±1.5 | **82.3±0.3** | |
| CIFAR10 | 1 | 0.02 | 14.4±2.0 | 21.5±1.2 | 21.5±1.3 | 13.5±1.2 | **28.3±0.5** | |
| | 10 | 0.2 | 26.0±1.2 | 31.6±0.7 | 14.7±0.9 | 23.3±1.0 | **44.9±0.5** | 84.8±0.1 |
| | 50 | 1 | 43.4±1.0 | 40.4±0.6 | 27.0±1.4 | 23.3±1.1 | **53.9±0.5** | |

Table 1: The performance comparison to coreset methods. This table shows the testing accuracies (%) of different methods on four datasets. ConvNet is used for training and testing. Img/Cls: image(s) per class, Ratio (%): the ratio of condensed images to whole training set.

learning (Snell et al., 2017; Vinyals et al., 2016; Gidaris & Komodakis, 2018) with $D$ duplicate blocks, and each block has a convolutional layer with $W$ ($3 \times 3$) filters, a normalization layer $N$, an activation layer $A$ and a pooling layer $P$, denoted as $[W, N, A, P] \times D$. The default ConvNet (unless specified otherwise) includes 3 blocks, each with 128 filters, followed by InstanceNorm (Ulyanov et al., 2016), ReLU and AvgPooling modules. The final block is followed by a linear classifier. We use Kaiming initialization (He et al., 2015) for network weights. The synthetic images can be initialized from Gaussian noise or randomly selected real training images. More details about the datasets, networks and hyper-parameters can be found in the supplementary.

The pipeline for dataset condensation has two stages: learning the condensed images (denoted as C) and training classifiers from scratch on them (denoted as T). Note that the model architectures used in two stages might be different. For the coreset baselines, the coreset is selected in the first stage. We investigate three settings: 1, 10 and 50 image/class learning, which means that the condensed set or coreset contains 1, 10 and 50 images per class respectively. Each method is run for 5 times, and 5 synthetic sets are generated in the first stage; each generated synthetic set is used to train 20 randomly initialized models in the second stage and evaluated on the test set, which amounts to evaluating 100 models in the second stage. In all experiments, we report the mean and standard deviation of these 100 testing results.

**Baselines.** We compare our method to four coreset baselines (Random, Herding, K-Center and Forgetting) and also to DD (Wang et al., 2018). In Random, the training samples are randomly selected as the coreset. Herding baseline, which selects closest samples to the cluster center, is based on (Welling, 2009) and used in (Rebuffi et al., 2017; Castro et al., 2018; Wu et al., 2019; Belouadah & Popescu, 2020). K-Center (Wolf, 2011; Sener & Savarese, 2018) picks multiple center points such that the largest distance between a data point and its nearest center is minimized. For Herding and K-Center, we use models trained on the whole dataset to extract features, compute $l_2$ distance to centers. Forgetting method (Toneva et al., 2019) selects the training samples which are easy to forget during training. We do not compare to GSS-Greedy (Aljundi et al., 2019), because it is also a similarity based greedy algorithm like K-Center, but GSS-Greedy trains an online learning model to measure the similarity of samples, which is different from general image classification problem. More detailed comparisons can be found in the supplementary.

**Comparison to coreset methods.** We first compare our method to the coreset baselines on MNIST, FashionMNIST, SVHN and CIFAR10 in Table 1 using the default ConvNet in classification accuracy. Whole dataset indicates training on the whole original set which serves as an approximate upper-bound performance. First we observe that our method outperforms all the baselines significantly and achieves a comparable result (98.8%) in case of 50 images per class to the upper bound (99.6%) in MNIST which uses two orders of magnitude more training images per class (6000). We also obtain promising results in FashionMNIST, however, the gap between our method and upper bound is bigger in SVHN and CIFAR10 which contain more diverse images with varying foregrounds and backgrounds. We also observe that, (i) the random selection baseline is competitive to other coreset methods in 10 and 50 images per class and (ii) herding method is on average the best coreset technique. We visualize the condensed images produced by our method under 1 image/class setting in Figure 2. Interestingly they are interpretable and look like "prototypes" of each class.

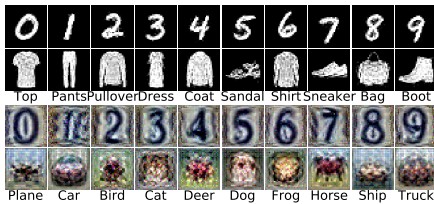

Figure 2: Visualization of condensed 1 image/class with ConvNet for MNIST, Fashion-MNIST, SVHN and CIFAR10.

| C\T | MLP | ConvNet | LeNet | AlexNet | VGG | ResNet |
|---|---|---|---|---|---|---|
| MLP | 70.5±1.2 | 63.9±6.5 | 77.3±5.8 | 70.9±11.6 | 53.2±7.0 | 80.9±3.6 |
| ConvNet | 69.6±1.6 | **91.7±0.5** | 85.3±1.8 | 85.1±3.0 | **83.4±1.8** | **90.0±0.8** |
| LeNet | 71.0±1.6 | 90.3±1.2 | 85.0±1.7 | 84.7±2.4 | 80.3±2.7 | 89.0±0.8 |
| AlexNet | 72.1±1.7 | 87.5±1.6 | 84.0±2.8 | 82.7±2.9 | 81.2±3.0 | 88.9±1.1 |
| VGG | 70.3±1.6 | 90.1±0.7 | 83.9±2.7 | 83.4±3.7 | 81.7±2.6 | 89.1±0.9 |
| ResNet | **73.6±1.2** | 91.6±0.5 | **86.4±1.5** | **85.4±1.9** | 83.4±2.4 | 89.4±0.9 |

Table 2: Cross-architecture performance in testing accuracy (%) for condensed 1 image/class in MNIST.

| Dataset | Img/Cls | DD | Ours | Whole Dataset |
|---|---|---|---|---|
| MNIST | 1 | - | 85.0±1.6 | 99.5±0.0 |
| | 10 | 79.5±8.1 | **93.9±0.6** | |
| CIFAR10 | 1 | - | 24.2±0.9 | 83.1±0.2 |
| | 10 | 36.8±1.2 | **39.1±1.2** | |

Table 3: Comparison to DD (Wang et al., 2018) in terms of testing accuracy (%).

| | Random | Herding | Ours | Early-stopping | Whole Dataset |
|---|---|---|---|---|---|
| Performance (%) | 76.2 | 76.2 | **84.5** | 84.5 | 85.9 |
| Correlation | -0.21 | -0.20 | **0.79** | 0.42 | 1.00 |
| Time cost (min) | **18.8** | **18.8** | **18.8** | **18.8** | 8604.3 |
| Storage (imgs) | $10^2$ | $10^2$ | $10^2$ | $10^4$ | $5 \times 10^4$ |

Table 4: Neural Architecture Search. Methods are compared in performance, ranking correlation, time and memory cost.

**Comparison to DD (Wang et al., 2018).** Unlike the setting in Table 1, DD (Wang et al., 2018) reports results only for 10 images per class on MNIST and CIFAR10 over LeNet and AlexCifarNet (a customized AlexNet). We strictly follow the experimental setting in (Wang et al., 2018), use the same architectures and report our and their original results in Table 3 for a fair comparison. Our method achieves significantly better performance than DD on both benchmarks; obtains 5% higher accuracy with only 1 synthetic sample per class than DD with 10 samples per class. In addition, our method obtains consistent results over multiple runs with a standard deviation of only 0.6% on MNIST, while DD's performance significantly vary over different runs (8.1%). Finally our method trains 2 times faster than DD and requires 50% less memory on CIFAR10 experiments. More detailed runtime and qualitative comparison can be found in the supplementary.

**Cross-architecture generalization.** Another key advantage of our method is that the condensed images learned using one architecture can be used to train another unseen one. Here we learn 1 condensed image per class for MNIST over a diverse set of networks including MLP, ConvNet (Gidaris & Komodakis, 2018), LeNet (LeCun et al., 1998), AlexNet (Krizhevsky et al., 2012), VGG-11 (Simonyan & Zisserman, 2014) and ResNet-18 (He et al., 2016) (see Table 2). Once the condensed sets are synthesized, we train every network on all the sets separately from scratch and evaluate their cross architecture performance in terms of classification accuracy on the MNIST test set. Table 2 shows that the condensed images, especially the ones that are trained with convolutional networks, perform well and are thus architecture generic. MLP generated images do not work well for training convolutional architectures which is possibly due to the mismatch between translation invariance properties of MLP and convolutional networks. Interestingly, MLP achieves better performance with convolutional network generated images than the MLP generated ones. The best results are obtained in most cases with ResNet generated images and ConvNet or ResNet as classifiers which is inline with their performances when trained on the original dataset.

**Number of condensed images.** We also study the test performance of a ConvNet trained on them for MNIST, FashionMNIST, SVHN and CIFAR10 for various number of condensed images per class in Figure 3 in absolute and relative terms – normalized by its upper-bound. Increasing the number of condensed images improves the accuracies in all benchmarks and further closes the gap with the upper-bound performance especially in MNIST and FashionMNIST, while the gap remains larger in SVHN and CIFAR10. In addition, our method outperforms the coreset method - Herding by a large margin in all cases.

**Activation, normalization & pooling.** We also study the effect of various activation (sigmoid, ReLU (Nair & Hinton, 2010; Zeiler et al., 2013), leaky ReLU (Maas et al., 2013)), pooling (max, average) and normalization functions (batch (Ioffe & Szegedy, 2015), group (Wu & He, 2018), layer (Ba et al., 2016), instance norm (Ulyanov et al., 2016)) and have the following observations: i) leaky ReLU over ReLU and average pooling over max pooling enable learning better condensed images, as they allow for denser gradient flow; ii) instance normalization obtains better classifica-

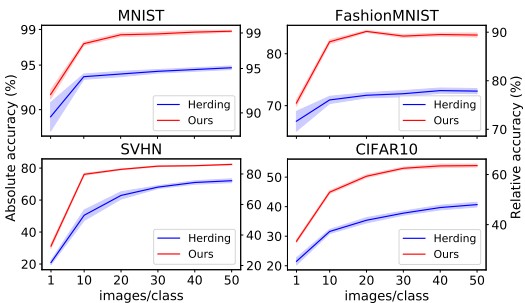
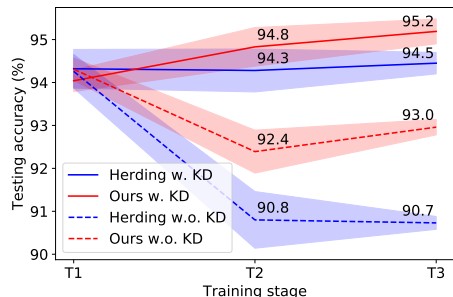

Figure 3: Absolute and relative testing accuracies for varying the number of condensed images/class for MNIST, FashionMNIST, SVHN and CIFAR10. The relative accuracy means the ratio compared to its upper-bound, *i.e.* training with the whole dataset.

Figure 4: Continual learning performance in accuracy (%). Herding denotes the original E2E (Castro et al., 2018). T1, T2, T3 are three learning stages. The performance at each stage is the mean testing accuracy on all learned tasks.

tion performance than its alternatives when used in the networks that are trained on a small set of condensed images. We refer to the supplementary for detailed results and discussion.

## 3.2 APPLICATIONS

**Continual Learning**  First we apply our method to a continual-learning scenario (Rebuffi et al., 2017; Castro et al., 2018) where new tasks are learned incrementally and the goal is to preserve the performance on the old tasks while learning the new ones. We build our model on E2E method in (Castro et al., 2018) that uses a limited budget rehearsal memory (we consider 10 images/class here) to keep representative samples from the old tasks and knowledge distillation (KD) to regularize the network's output w.r.t. to previous predictions. We replace its sample selection mechanism (herding) with ours such that a set of condensed images are generated and stored in the memory, keep the rest of the model same and evaluate this model on the task-incremental learning problem on the digit recognition datasets, SVHN (Netzer et al., 2011), MNIST (LeCun et al., 1998) and USPS (Hull, 1994) in the same order. MNIST and USPS images are reshaped to $32 \times 32$ RGB images.

We compare our method to E2E (Castro et al., 2018), depicted as herding in Figure 4, with and without KD regularization. The experiment contains 3 incremental training stages (SVHN→MNIST→USPS) and testing accuracies are computed by averaging over the test sets of the previous and current tasks after each stage. The desired outcome is to obtain high mean classification accuracy at T3. The results indicate that the condensed images are more *data-efficient* than the ones sampled by herding and thus our method outperforms E2E in both settings, while by a larger margin (2.3% at T3) when KD is not employed.

**Neural Architecture Search.**  Here we explore the use of our method in a simple neural architecture search (NAS) experiment on CIFAR10 which typically requires expensive training of numerous architectures multiple times on the whole training set and picking the best performing ones on a validation set. Our goal is to verify that our condensed images can be used to efficiently train multiple networks to identify the best network. To this end, we construct the search space of 720 ConvNets as described in Section 3.1 by varying hyper-parameters $W$, $N$, $A$, $P$, $D$ over an uniform grid (see supplementary for more details), train them for 100 epochs on three small proxy datasets (10 images/class) that are obtained with Random sampling, Herding and our method. Note that we train the condensed images for once only with the default ConvNet architecture and use them to train all kinds of architectures. We also compare to early-stopping (Li & Talwalkar, 2020) in which the model is trained on whole training set but with the same number of training iterations as the one required for the small proxy datasets, in other words, for the same amount of computations.

Table 4 depicts i) the average test performance of the best selected model over 5 runs when trained on the whole dataset, ii) Spearman's rank correlation coefficient between the validation accuracies obtained by training the selected top 10 models on the proxy dataset and whole dataset, iii) time for training 720 architectures on a NVIDIA GTX1080-Ti GPU, and iv) memory print of the training images. Our method achieves the highest testing performance (84.5%) and performance correlation (0.79), meanwhile significantly decreases the the searching time (from 8604.3 to 18.8 minutes) and storage space (from $5 \times 10^4$ to $1 \times 10^2$ images) compared to whole-dataset training. The competitive early-stopping baseline achieves on par performance for the best performing model with

ours, however, the rank correlation (0.42) of top 10 models is significantly lower than ours (0.79) which indicates unreliable correlation of performances between early-stopping and whole-dataset training. Furthermore, early-stopping needs 100 times as many training images as ours needs. Note that the training time for synthetic images is around 50 minutes (for $K = 500$) which is one time off and negligible cost when training thousands even millions of candidate architectures in NAS.

## 4    CONCLUSION

In this paper, we propose a dataset condensation method that learns to synthesize a small set of informative images. We show that these images are significantly more data-efficient than the same number of original images and the ones produced by the previous method, and they are not architecture dependent, can be used to train different deep networks. Once trained, they can be used to lower the memory print of datasets and efficiently train numerous networks which are crucial in continual learning and neural architecture search respectively. For future work, we plan to explore the use of condensed images in more diverse and thus challenging datasets like ImageNet (Deng et al., 2009) that contain higher resolution images with larger variations in appearance and pose of objects, background.

**Acknowledgment.**    This work is funded by China Scholarship Council 201806010331 and the EPSRC programme grant Visual AI EP/T028572/1. We thank Iain Murray and Oisin Mac Aodha for their valuable feedback.

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

## A    IMPLEMENTATION DETAILS

In this part, we explain the implementation details for the dataset condensation, continual learning and neural architecture search experiments.

**Dataset condensation.**    The presented experiments involve tuning of six hyperparameters – the number of outer-loop $K$ and inner-loop steps $T$, learning rates $\eta_{\mathcal{S}}$ and number of optimization steps $\varsigma_{\mathcal{S}}$ for the condensed samples, learning rates $\eta_{\theta}$ and number of optimization steps $\varsigma_{\theta}$ for the model weights. In all experiments, we set $K = 1000$, $\eta_{\mathcal{S}} = 0.1$, $\eta_{\theta} = 0.01$, $\varsigma_{\mathcal{S}} = 1$ and employ Stochastic Gradient Descent (SGD) as the optimizer. The only exception is that we set $\eta_{\mathcal{S}}$ to $0.01$ for synthesizing data with MLP in cross-architecture experiments (Table 2), as MLP requires a slightly different treatment. Note that while $K$ is the maximum number of outer-loop steps, the optimization can early-stop automatically if it converges before $K$ steps. For the remaining hyperparameters, we use different sets for 1, 10 and 50 image(s)/class learning. We set $T = 1$, $\varsigma_{\theta} = 1$ for 1 image/class, $T = 10$, $\varsigma_{\theta} = 50$ for 10 images/class, $T = 50$, $\varsigma_{\theta} = 10$ for 50 images/class learning. Note that when $T = 1$, it is not required to update the model parameters (Step 9 in Algorithm 1), as this model is not further used. For those experiments where more than 10 images/class are synthesized, we set $T$ to be the same number as the synthetic images per class and $\varsigma_{\theta} = 500/T$, $e.g.$ $T = 20$, $\varsigma_{\theta} = 25$ for 20 images/class learning. The ablation study on hyper-parameters are given in Appendix B which shows that our method is not sensitive to varying hyper-parameters.

We do separate-class mini-batch sampling for Step 6 in Algorithm 1. Specifically, we sample a mini-batch pair $B_c^{\mathcal{T}}$ and $B_c^{\mathcal{S}}$ that contain real and synthetic images from the same class $c$ at each inner iteration. Then, the matching loss for each class is computed with the sampled mini-batch pair and used to update corresponding synthetic images $\mathcal{S}_c$ by back-propogation (Step 7 and 8). This is repeated separately (or parallelly given enough computational resources) for every class. Training as such is not slower than using mixed-class batches. Although our method still works well when we randomly sample the real and synthetic mini-batches with mixed labels, we found that separate-class strategy is faster to train as matching gradients w.r.t. data from single class is easier compared to those of multiple classes. In experiments, we randomly sample 256 real images of a class as a mini-batch to calculate the mean gradient and match it with the mean gradient that is averaged over all synthetic samples with the same class label. The performance is not sensitive to the size of real-image mini-batch if it is greater than 64.

In all experiments, we use the standard train/test splits of the datasets – the train/test statistics are shown in Table T5. We apply data augmentation (crop, scale and rotate) only for experiments (coreset methods and ours) on MNIST. The only exception is that we also use data augmentation when compared to DD (Wang et al., 2018) on CIFAR10 with AlexCifarNet, and data augmentation is also used in (Wang et al., 2018). For initialization of condensed images, we tried both Gaussian noise and randomly selected real training images, and obtained overall comparable performances in different settings and datasets. Then, we used Gaussian noise for initialization in experiments.

|  | USPS | MNIST | FashionMNIST | SVHN | CIFAR10 | CIFAR100 |
|---|---|---|---|---|---|---|
| Train | 7,291 | 60,000 | 60,000 | 73,257 | 50,000 | 50,000 |
| Test | 2,007 | 10,000 | 10,000 | 26,032 | 10,000 | 10,000 |

Table T5: Train/test statistics for USPS, MNIST, FashionMNIST, SVHN, CIFAR10 and CIFAR100 datasets.

In the first stage – while training the condensed images –, we use Batch Normalization in the VGG and ResNet networks. For reliable estimation of the running mean and variance, we sample many real training data to estimate the running mean and variance and then freeze them ahead of Step 7. In the second stage – while training a deep network on the condensed set –, we replace Batch Normalization layers with Instance Normalization in VGG and ResNet, due to the fact that the batch statistics are not reliable when training networks with few condensed images. Another minor modification that we apply to the standard network ResNet architecture in the first stage is replacing the strided convolutions where $stride = 2$ with convolutional layers where $stride = 1$ coupled with an average pooling layer. We observe that this change enables more detailed (per pixel) gradients w.r.t. the condensed images and leads to better condensed images.

**Continual learning.**    In this experiment, we focus on a task-incremental learning on SVHN, MNIST and USPS with the given order. The three tasks share the same label space, however have

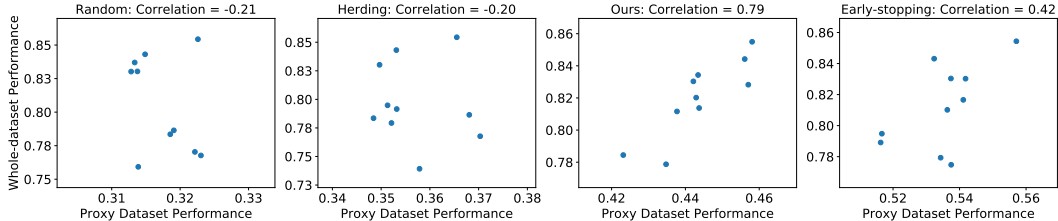

Figure F5: The performance correlation between the training on proxy dataset and whole-dataset. For each proxy dataset, the best 10 models are selected based on validation set performance. In the figure, each point represents an architecture.

| C\T | Sigmoid | ReLu | LeakyReLu |
|---|---|---|---|
| Sigmoid | 86.7±0.7 | 91.2±0.6 | 91.2±0.6 |
| ReLu | 86.1±0.9 | 91.7±0.5 | 91.7±0.5 |
| LeakyReLu | 86.3±0.9 | 91.7±0.5 | 91.7±0.4 |

Table T6: Cross-activation experiments in accuracy (%) for 1 condensed image/class in MNIST.

| C\T | None | MaxPooling | AvgPooling |
|---|---|---|---|
| None | 78.7±3.0 | 80.8±3.5 | 88.3±1.0 |
| MaxPooling | 81.2±2.8 | 89.5±1.1 | 91.1±0.6 |
| Avgpooing | 81.8±2.9 | 90.2±0.8 | 91.7±0.5 |

Table T7: Cross-pooling experiments in accuracy (%) for 1 condensed image/class in MNIST.

significantly different image statistics. The images of the three datasets are reshaped to $32 \times 32$ RGB size for standardization. We use the standard splits for training sets and randomly sample 2,000 test images for each datasets to obtain a balanced evaluation over three datasets. Thus each model is tested on a growing test set with 2,000, 4,000 and 6,000 images at the three stages respectively. We use the default ConvNet in this experiment and set the weight of distillation loss to 1.0 and the temperature to 2. We run 5,000 and 500 iterations for training and balanced finetuning as in (Castro et al., 2018) with the learning rates 0.01 and 0.001 respectively. We run 5 experiments and report the mean and standard variance in Figure 4.

**Neural Architecture Search.** To construct the searching space of 720 ConvNets, we vary hyper-parameters $W \in \{32, 64, 128, 256\}$, $D \in \{1, 2, 3, 4\}$, $N \in \{$None, BatchNorm, LayerNorm, InstanceNorm, GroupNorm$\}$, $A \in \{$Sigmoid, ReLu, LeakyReLu$\}$, $P \in \{$None, MaxPooling, AvgPooling$\}$. We randomly sample 5,000 images from the 50,000 training images in CIFAR10 as the validation set. Every candidate ConvNet is trained with the proxy dataset, and then evaluated on the validation set. These candidate ConvNets are ranked by the validation performance. 10 architectures with top validation accuracies are selected to calculate Spearman's rank correlation coefficient, because the best model that we want will come from the top 10 architectures. We train each ConvNet for 5 times to get averaged validation and testing accuracies.

We visualize the performance correlation for different proxy datasets in Figure F5. Obviously, the condensed proxy dataset produced by our method achieves the highest performance correlation (0.79) which significantly higher than early-stopping (0.42). It means our method can produce more reliable results for NAS.

## B  FURTHER ANALYSIS

Next we provide additional results on ablative studies over various deep network layers including activation, pooling and normalization functions and also over depth and width of deep network architecture. We also study the selection of hyper-parameters and the gradient distance metric. An additional qualitative analysis on the learned condensed images is also given.

**Ablation study on activation functions.** Here we study the use of three activation functions – Sigmoid, ReLU, LeakyReLu (negative slope is set to 0.01) – in two stages, when training condensed images (denoted as C) and when training a ConvNet from scratch on the learned condensed images (denoted as T). The experiments are conducted in MNIST dataset for 1 condensed image/class setting. Table T6 shows that all three activation functions are good for the first stage while generating good condensed images, however, Sigmoid performs poor in the second stage while learning a classifier on the condensed images – its testing accuracies are lower than ReLu and LeakyReLu by around $5\%$. This suggests that ReLU can provide sufficiently informative gradients for learning condensed images, though the gradient of ReLU w.r.t. to its input is typically sparse.

| C\T | None | BatchNorm | LayerNorm | InstanceNorm | GroupNorm |
|---|---|---|---|---|---|
| None | 79.0±2.2 | 80.8±2.0 | 85.8±1.7 | 90.7±0.7 | 85.9±1.7 |
| BatchNorm | 78.6±2.1 | 80.7±1.8 | 85.7±1.6 | 90.9±0.6 | 85.9±1.5 |
| LayerNorm | 81.2±1.8 | 78.6±3.0 | 87.4±1.3 | 90.7±0.7 | 87.3±1.4 |
| InstanceNorm | 72.9±7.1 | 56.7±6.5 | 82.7±5.3 | 91.7±0.5 | 84.3±4.2 |
| GroupNorm | 79.5±2.1 | 81.8±2.3 | 87.3±1.2 | 91.6±0.5 | 87.2±1.2 |

Table T8: Cross-normalization experiments in accuracy (%) for 1 condensed image/class in MNIST.

| C\T | 1 | 2 | 3 | 4 |
|---|---|---|---|---|
| 1 | 61.3±3.5 | 78.2±3.0 | 77.1±4.0 | 76.4±3.5 |
| 2 | 78.3±2.3 | 89.0±0.8 | 91.0±0.6 | 89.4±0.8 |
| 3 | 81.6±1.5 | 89.8±0.8 | 91.7±0.5 | 90.4±0.6 |
| 4 | 82.5±1.3 | 89.9±0.8 | 91.9±0.5 | 90.6±0.4 |

Table T9: Cross-depth performance in accuracy (%) for 1 condensed image/class in MNIST.

| C\T | 32 | 64 | 128 | 256 |
|---|---|---|---|---|
| 32 | 90.6±0.8 | 91.4±0.5 | 91.5±0.5 | 91.3±0.6 |
| 64 | 91.0±0.8 | 91.6±0.6 | 91.8±0.5 | 91.4±0.6 |
| 128 | 90.8±0.7 | 91.5±0.6 | 91.7±0.5 | 91.2±0.7 |
| 256 | 91.0±0.7 | 91.6±0.6 | 91.7±0.5 | 91.4±0.5 |

Table T10: Cross-width performance in accuracy (%) for 1 condensed image/class in MNIST.

**Ablation study on pooling functions.** Next we investigate the performance of two pooling functions – average pooling and max pooling – also no pooling for 1 image/class dataset condensation with ConvNet in MNIST in terms of classification accuracy. Table T7 shows that max and average pooling both perform significantly better than no pooling (None) when they are used in the second stage. When the condensed samples are trained and tested on models with average pooling, the best testing accuracy ($91.7 \pm 0.5\%$) is obtained, possibly, because average pooling provides more informative and smooth gradients for the whole image rather than only for its discriminative parts.

**Ablation study on normalization functions.** Next we study the performance of four normalization options – No normalization, Batch (Ioffe & Szegedy, 2015), Layer (Ba et al., 2016), Instance (Ulyanov et al., 2016) and Group Normalization (Wu & He, 2018) (number of groups is set to be four) – for 1 image/class dataset condensation with ConvNet architecture in MNIST classification accuracy. Table T8 shows that the normalization layer has little influence for learning the condensed set, while the choice of normalization layer is important for training networks on the condensed set. LayerNorm and GroupNorm have similar performance, and InstanceNorm is the best choice for training a model on condensed images. BatchNorm obtains lower performance which is similar to None (no normalization), as it is known to perform poorly when training models on few condensed samples as also observed in (Wu & He, 2018). Note that Batch Normalization does not allow for a stable training in the first stage (C); thus we replace its running mean and variance for each batch with those of randomly sampled real training images.

**Ablation study on network depth and width.** Here we study the effect of network depth and width for 1 image/class dataset condensation with ConvNet architecture in MNIST in terms of classification accuracy. To this end we conduct multiple experiments by varying the depth and width of the networks that are used to train condensed synthetic images and that are trained to classify testing data in ConvNet architecture and report the results in Table T9 and Table T10. In Table T9, we observe that deeper ConvNets with more blocks generate better condensed images that results in better classification performance when a network is trained on them, while ConvNet with 3 blocks performs best as classifier. Interestingly, Table T10 shows that the best results are obtained with the classifier that has 128 filters at each block, while network width (number of filters at each block) in generation has little overall impact on the final classification performance.

**Ablation study on hyper-parameters.** Our performance is not sensitive to hyper-parameter selection. The testing accuracy for various $K$ and $T$, when learning 10 images/class condensed sets, is depicted in Figure F6. The results show that the optimum $K$ and $T$ are around similar values across all datasets. Thus we simply set $K$ to 1000 and $T$ to 10 for all datasets. Similarly, for the remaining ones including learning rate, weight decay, we use a single set of hyperparameters that are observed to work well for all datasets and architectures in our preliminary experiments.

**Ablation study on gradient distance metric.** To prove the effectiveness and robustness of the proposed distance metric for gradients (or weights), we compare to the traditional ones (Lopez-Paz et al., 2017; Aljundi et al., 2019; Zhu et al., 2019) which vectorize and concatenate the whole gradient, $\mathbf{G}^{\mathcal{T}}, \mathbf{G}^{\mathcal{S}} \in \mathbb{R}^{D}$, and compute the squared Euclidean distance $\|\mathbf{G}^{\mathcal{T}} - \mathbf{G}^{\mathcal{S}}\|^2$ and the Cosine distance $1 - \cos(\mathbf{G}^{\mathcal{T}}, \mathbf{G}^{\mathcal{S}})$, where $D$ is the number of all network parameters. We do 1

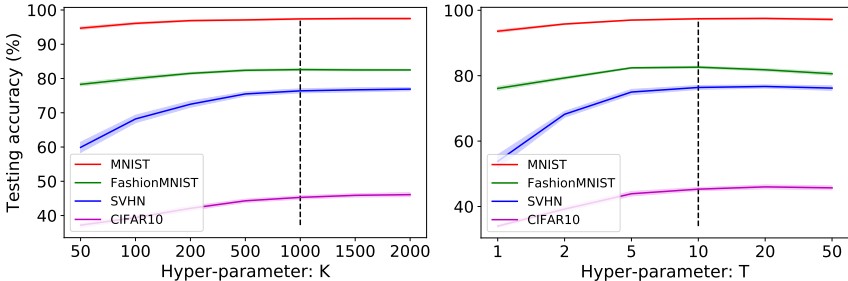

Figure F6: Ablation study on the hyper-parameters $K$ and $T$ when learning 10 images/class condensed sets.

image/class learning experiment on MNIST with different architectures. For simplicity, the synthetic images are learned and tested on the same architecture in this experiment. Table T11 shows that the proposed gradient distance metric remarkably outperforms others on complex architectures (*e.g.* LeNet, AlexNet, VGG and ResNet) and achieves the best performances in most settings, which means it is more effective and robust than the traditional ones. Note that we set $\eta_{\mathcal{S}} = 0.1$ for MLP-Euclidean and MLP-Cosine because it works better than $\eta_{\mathcal{S}} = 0.01$.

| | MLP | ConvNet | LeNet | AlexNet | VGG | ResNet |
|---|---|---|---|---|---|---|
| Euclidean | 69.3±0.9 | **92.7±0.3** | 65.0±5.1 | 66.2±5.6 | 57.1±7.0 | 68.0±5.2 |
| Cosine | 45.2±3.6 | 69.2±2.7 | 61.1±8.2 | 58.3±4.1 | 55.0±5.0 | 68.8±7.8 |
| Ours | **70.5±1.2** | 91.7±0.5 | **85.0±1.7** | **82.7±2.9** | **81.7±2.6** | **89.4±0.9** |

Table T11: Ablation study on different gradient distance metrics. Obviously, the proposed distance metric is more effective and robust. Euclidean: squared Euclidean distance, Cosine: Cosine distance.

**Further qualitative analysis** We first depict the condensed images that are learned on MNIST, FashionMNIST, SVHN and CIFAR10 datasets in one experiment using the default ConvNet in 10 images/class setting in Figure F7. It is interesting that the 10 images/class results in Figure F7 are diverse which cover the main variations, while the condensed images for 1 image/class setting (see Figure 2) look like the "prototype" of each class. For example, in Figure F7 (a), the ten images of "four" indicate ten different styles. The ten "bag" images in Figure F7 (b) are significantly different from each other, similarly "wallet" (1st row), "shopping bag" (3rd row), "handbag" (8th row) and "schoolbag" (10th row). Figure F7 (c) also shows the diverse house numbers with different shapes, colors and shadows. Besides, different poses of a "horse" have been learned in Figure F7 (d).

## C  COMPARISON TO MORE BASELINES

**Optimal random selection.** One interesting and strong baseline is Optimal Random Selection (ORS) that we implement random selection experiments for 1,000 times and pick the best ones. Table T12 presents the performance comparison to the selected Top 1000 (all), Top 100 and Top 10 coresets. These optimal coresets are selected by ranking their performance. Obviously, the condensed set generated by our method surpasses the selected Top 10 of 1000 coresets with a large margin on all four datasets.

**Generative model.** We also compare to the popular generative model, *namely*, Conditional Generative Adversarial Networks (cGAN) (Mirza & Osindero, 2014). The generator has two blocks which consists of the Up-sampling (scale_factor=2), Convolution (stride=1), BatchNorm and LeakyReLu layers. The discriminator has three blocks which consists of Convolution (stride=2), BatchNorm and LeakyReLu layers. In additional to the random noise, we also input the class label as the condition. We generate 1 and 10 images per class for each dataset with random noise. Table T12 shows that the images produced by cGAN have similar performances to those randomly selected coresets (*i.e.* Top 1000). It is reasonable, because the aim of cGAN is to generate real-look images. In contrast, our method aims to generate images that can train deep neural networks efficiently.

**Analysis of coreset performances** We find that K-Center (Wolf, 2011; Sener & Savarese, 2018) and Forgetting (Toneva et al., 2019) don't work as well as other general coreset methods, namely

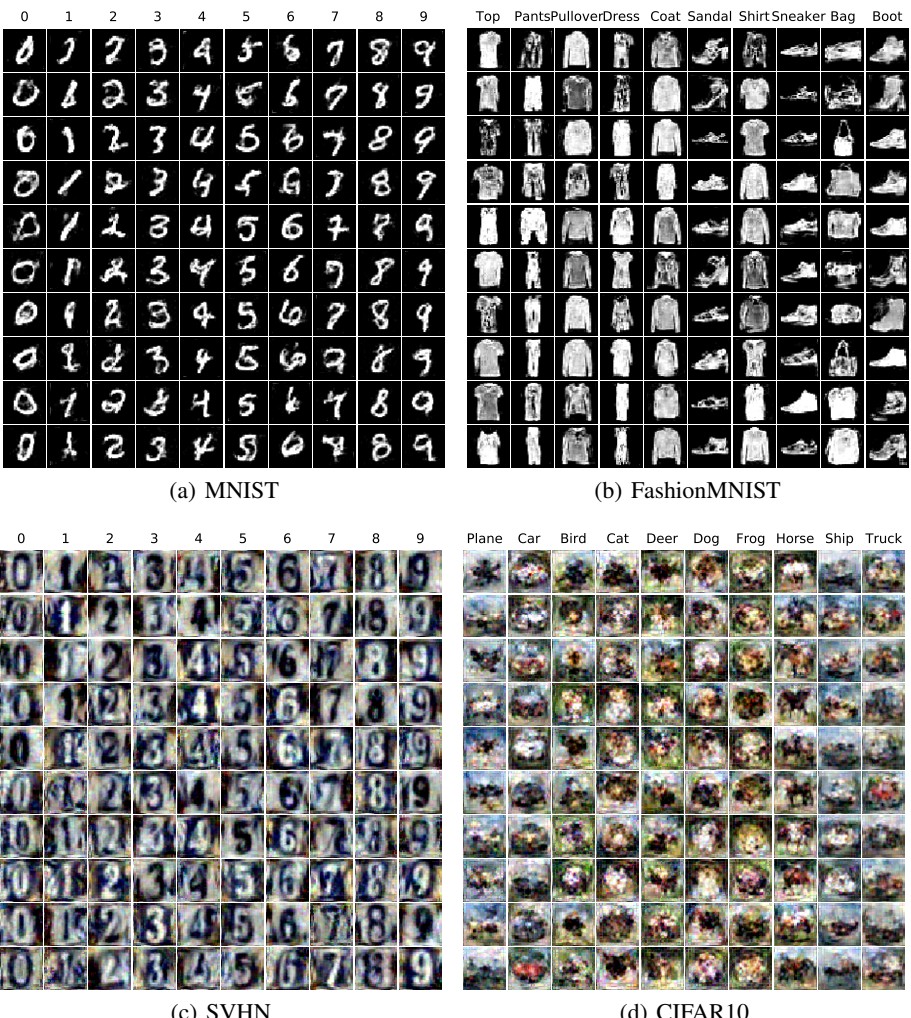

Figure F7: The synthetic images for MNIST, FashionMNIST, SVHN and CIFAR10 produced by our method with ConvNet under 10 images/class setting.

| | Img/Cls | Ratio % | Optimal Random Selection | | | cGAN | Ours | Whole Dataset |
| | | | Top 1000 | Top 100 | Top 10 | | | |
|---|---|---|---|---|---|---|---|---|
| MNIST | 1 | 0.017 | 64.3±6.1 | 74.4±1.8 | 78.2±1.7 | 64.0±3.2 | **91.7±0.5** | 99.6±0.0 |
| | 10 | 0.17 | 94.8±0.7 | 96.0±0.2 | 96.4±0.1 | 94.9±0.6 | **97.4±0.2** | |
| FashionMNIST | 1 | 0.017 | 51.3±5.4 | 59.6±1.3 | 62.4±0.9 | 51.1±0.8 | **70.5±0.6** | 93.5±0.1 |
| | 10 | 0.17 | 73.8±1.6 | 76.4±0.6 | 77.6±0.2 | 73.9±0.7 | **82.3±0.4** | |
| SVHN | 1 | 0.014 | 14.3±2.1 | 18.1±0.9 | 19.9±0.2 | 16.1±0.9 | **31.2±1.4** | 95.4±0.1 |
| | 10 | 0.14 | 34.6±3.2 | 40.3±1.3 | 42.9±0.9 | 33.9±1.1 | **76.1±0.6** | |
| CIFAR10 | 1 | 0.02 | 15.0±2.0 | 18.5±0.8 | 20.1±0.5 | 16.3±1.4 | **28.3±0.5** | 84.8±0.1 |
| | 10 | 0.2 | 27.1±1.6 | 29.8±0.7 | 31.4±0.2 | 27.9±1.1 | **44.9±0.5** | |

Table T12: The performance comparison to optimal random selection (ORS) and conditional generative adversarial networks (cGAN) baselines. This table shows the testing accuracies (%) of different methods on four datasets. ConvNet is used for training and testing. Img/Cls: image(s) per class, Ratio (%): the ratio of condensed images to whole training set. Top 1000, Top 100 and Top 10 means the selected 1000, 100 and 10 optimal coresets by ranking their performances.

| | Img/Cls | Ratio % | Core-set Selection | | | | LD[†] | Ours | Whole Dataset |
|---|---|---|---|---|---|---|---|---|---|
| | | | Random | Herding | K-Center | Forgetting | | | |
| CIFAR100 | 1 | 0.2 | 4.2±0.3 | 8.4±0.3 | 8.3±0.3 | 3.5±0.3 | 11.5±0.4 | **12.8±0.3** | 56.2±0.3 |
| | 10 | 2 | 14.6±0.5 | 17.3±0.3 | 7.1±0.3 | 9.8±0.2 | - | **25.2±0.3** | |

Table T13: The performance comparison on CIFAR100. This table shows the testing accuracies (%) of different methods. ConvNet is used for training and testing except that LD[†] uses AlexNet. Img/Cls: image(s) per class, Ratio (%): the ratio of condensed images to whole training set.

| Method | MLP | ConvNet | LeNet | AlexNet | VGG | ResNet |
|---|---|---|---|---|---|---|
| DD | 72.7±2.8 | 77.6±2.9 | 79.5±8.1 | 51.3±19.9 | 11.4±2.6 | 63.6±12.7 |
| Ours | **83.0±2.5** | **92.9±0.5** | **93.9±0.6** | **90.6±1.9** | **92.9±0.5** | **94.5±0.4** |

Table T14: Generalization ability comparison to DD. The 10 condensed images per class are trained with LeNet, and tested on various architectures. It shows that condensed images generated by our method have better generalization ability.

Random and Herding (Rebuffi et al., 2017), in this experimental setting. After analyzing the algorithms and coresets, we find two main reasons. 1) K-Center and Forgetting are not designed for training deep networks from scratch, instead they are for active learning and continual learning respectively. 2) The two algorithms both tend to select "hard" samples which are often outliers when only a small number of images are selected. These outliers confuse the training, which results in worse performance. Specifically, the first sample per class in K-Center coreset is initialized by selecting the one closest to each class center. The later ones selected by the greedy criterion that pursues maximum coverage are often outliers which confuse the training.

**Performance on CIFAR100.** We supplement the performance comparison on CIFAR100 dataset which includes 10 times as many classes as other benchmarks. More classes while fewer images per class makes CIFAR100 significantly more challenging than other datasets. We use the same set of hyper-parameters for CIFAR100 as other datasets. Table T13 depicts the performances of coreset selection methods, Label Distillation (LD) Bohdal et al. (2020) and ours. Our method achieves 12.8% and 25.2% testing accuracies on CIFAR100 when learning 1 and 10 images per class, which are the best compared with others.

## D    FURTHER COMPARISON TO DD (WANG ET AL., 2018)

Next we compare our method to DD (Wang et al., 2018) first quantitatively in terms of cross-architecture generalization, then qualitatively in terms of synthetic image quality, and finally in terms of computational load for training synthetic images. Note that we use the original source code to obtain the results for DD that is provided by the authors of DD in the experiments.

**Generalization ability comparison.** Here we compare the generalization ability across different deep network architectures to DD. To this end, we use the synthesized 10 images/class data learned with LeNet on MNIST to train MLP, ConvNet, LeNet, AlexNet, VGG11 and ResNet18 and report the results in Table T14. We see that that the condensed set produced by our method achieves good classification performances with all architectures, while the synthetic set produced by DD perform poorly when used to trained some architectures, *e.g.* AlexNet, VGG and ResNet. Note that DD generates learning rates to be used in every training step in addition to the synthetic data. This is in contrast to our method which does not learn learning rates for specific training steps. Although the tied learning rates improve the performance of DD while training and testing on the same architecture, they will hinder the generalization to unseen architectures.

| Method | Dataset | Architecture | Memory (MB) | Time (min) | Test Acc. |
|---|---|---|---|---|---|
| DD | MNIST | LeNet | 785 | 160 | 79.5±8.1 |
| Ours | MNIST | LeNet | 653 | 46 | 93.9±0.6 |
| DD | CIFAR10 | AlexCifarNet | 3211 | 214 | 36.8±1.2 |
| Ours | CIFAR10 | AlexCifarNet | 1445 | 105 | 39.1±1.2 |

Table T15: Time and memory use for training DD and our method in 10 images/class setting.

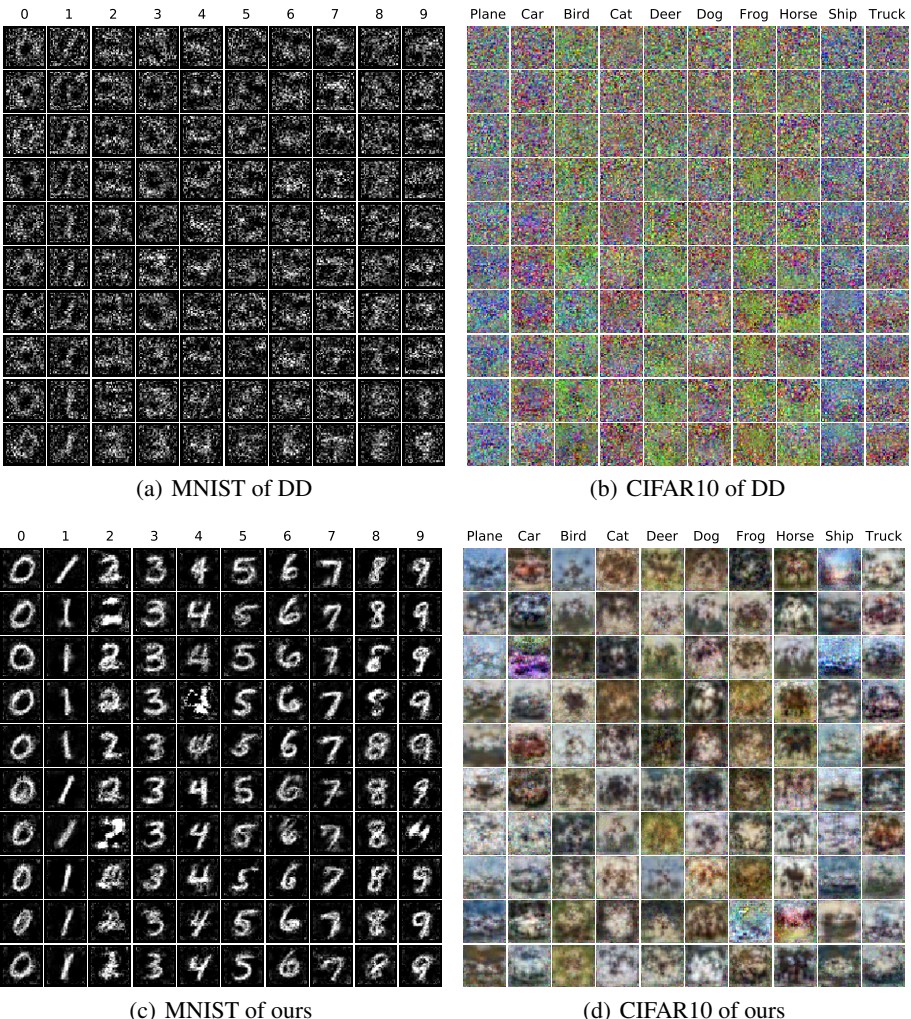

Figure F8: Qualitative comparison between the condensed images produced by DD and ours under 10 images/class setting. LeNet and AlexCifarNet are utilized for MNIST and CIFAR10 respectively.

**Qualitative comparison.** We also provide a qualitative comparison to to DD in terms of image quality in Figure F8. Note that both of the synthetic sets are trained with LeNet on MNIST and AlexCifarNet on CIFAR10. Our method produces more interpretable and realistic images than DD, although it is not our goal. The MNIST images produced by DD are noisy, and the CIFAR10 images produced by DD do not show any clear structure of the corresponding class. In contrast, the MNIST and CIFAR10 images produced by our method are both visually meaningful and diverse.

**Training memory and time.** One advantage of our method is that we decouple the model weights from its previous states in training, while DD requires to maintain the recursive computation graph which is not scalable to large models and inner-loop optimizers with many steps. Hence, our method requires less training time and memory cost. We compare the training time and memory cost required by DD and our method with one NVIDIA GTX1080-Ti GPU. Table T15 shows that our method requires significantly less memory and training time than DD and provides an approximation reduction of 17% and 55% in memory and 71% and 51% in train time to learn MNIST and CIFAR10 datasets respectively. Furthermore, our training time and memory cost can be significantly decreased by using smaller hyper-parameters, *e.g.* $K$, $T$ and the batch size of sampled real images, with a slight performance decline (refer to Figure F6).

# E    EXTENDED RELATED WORK

**Variations of Dataset Distillation.**    There exists recent work that extends Dataset Distillation (Wang et al., 2018). For example, (Sucholutsky & Schonlau, 2019; Bohdal et al., 2020) aim to improve DD by learning soft labels with/without synthetic images. (Such et al., 2020) utilizes a generator to synthesize images instead of directly updating image pixels. However, the reported quantitative and qualitative improvements over DD are minor compared to our improvements. In addition, none of these methods have thoroughly verified the cross-architecture generalization ability of the synthetic images.

**Zero-shot Knowledge Distillation.**    Recent zero-shot KD methods (Lopes et al., 2017; Nayak et al., 2019) aim to perform KD from a trained model in the absence of training data by generating synthetic data as the intermediate production to further use. Unlike them, our method does not require pretrained teacher models to provide the knowledge, *i.e*. to obtain the features and labels.

**Data Privacy & Federated Learning.**    Synthetic dataset is also a promising solution to protecting data privacy and enabling safe federated learning. There exists some work that uses synthetic dataset to protect the privacy of medical dataset (Li et al., 2020) and reduce the communication rounds in federated learning (Zhou et al., 2020). Although transmitting model weights or gradients (Zhu et al., 2019; Zhao et al., 2020) may increase the transmission security, the huge parameters of modern deep neural networks are prohibitive to transmit frequently. In contrast, transmitting small-scale synthetic dataset between clients and server is low-cost (Goetz & Tewari, 2020).

