# OpenReview forum: "Dataset Condensation with Gradient Matching"
_ICLR.cc/2021/Conference — ICLR 2021 Oral_

### Official Review · AnonReviewer2 · 2020-10-25
**Good paper with extensive experiments**

**Rating:** 8
**Confidence:** 3

**Review:**

Summary: This paper tackles the challenging dataset condensation problem. The goal is to learn to synthesize a small dataset, so that a neural network trained on the small synthetic dataset can have similar performance as a network trained on the full dataset. The proposed method tackles the problem by gradient matching. The proposed method achieves state-of-the-art performance, and shows promising results on two other downstream tasks, continual learning and neural architecture search.

Strength:
+ Comparing to existing approaches, the proposed method is efficient and effective, achieving the state-of-the-art performance
+ The authors use the synthetic dataset for two other downstream tasks and achieve promising results
+ The authors conduct extensive experiments to study and analyze the proposed method
+ The synthetic dataset trained on one architecture can be also used to train any other networks with different architectures, which makes to method more applicable

Weakness:
- In Section 2.3 “Gradient matching loss”, the authors claim that the proposed distance metric is “a better distance”. It is probably better to use experiments or results to support this claim.

Other comments:
* Currently, the loss for the target task is cross-entropy loss only (Figure 1(b)). I wonder if this method can be used for other loss functions as well. Also, I wonder if the proposed method can be used for self-supervised tasks as well. Can the authors comment on these?
* It seems that the authors do not have any special designs to make the condensed synthetic dataset cross-architecture generalizable. I wonder why the proposed method has such a good cross-architecture generalization.
* In Figure 3, it seems that the performance on CIFAR10 saturates quickly as the number of images per class grows. Just curious, will the relative performance eventually reach 80~90% as in other datasets? If so, then what is the data ratio?

---- Post-rebuttal comments----

Thanks for the response. After reading other reviews' comments and the rebuttal, I think this paper is in a good shape now. Thus, I am willing to increase my score to 8 and recommend acceptance.

---

> ### Author Response · Authors · 2020-11-16
> **Response to AnonReviewer2**
>
> We thank the reviewer for his/her valuable comments and time.
>
> A1 "Experimental justification for the gradient matching loss": We agree with the reviewer that it is important to justify the decision. To this end, we compare the proposed distance metric to standard Euclidean distance and cosine distance after vectorising gradient tensors each layer and concatenating them as a single vector. By using each distance, we learn one condensed image per class on MNIST dataset and use them to train the same architecture from scratch. We repeat this experiment on different architectures (MLP, ConvNet, LeNet, AlexNet, VGG11 and ResNet18) and report their test accuracies below. The results show that the proposed distance metric obtains substantially better classification results than two baselines. We have added this ablation study in the supplementary material - Further Analysis - Ablation study on gradient distance metric. For more details, please refer to the revised paper.
>
> |           | MLP      | ConvNet  | LeNet    | AlexNet  | VGG      | ResNet   |
> |-----------------|----------------|----------------|----------------|----------------|----------------|----------------|
> | Euclidean | 69.3±0.9 | 92.7±0.3 | 65.0±5.1 | 66.2±5.6 | 57.1±7.0 | 68.0±5.2 |
> | Cosine    | 45.2±3.6 | 69.2±2.7 | 61.1±8.2 | 58.3±4.1 | 55.0±5.0 | 68.8±7.8 |
> | Ours      | 70.5±1.2 | 91.7±0.5 | 85.0±1.7 | 82.7±2.9 | 81.7±2.6 | 89.4±0.9 |
>
> A2 "Other loss function than cross-entropy": Although we have only trained our images on classification problems with cross-entropy loss, we cannot see any obvious reason that it should not work on a different loss function such as L1/L2 loss, given that it is differentiable. In future, we plan to test it on a regression problem such as facial landmark regression.
>
> A3 "Self-supervised learning": Our method could be used in self-supervised learning problems such as estimating rotation of an image (Gidaris et al 2018 ICLR) without any major modification. For more recent ones such as Chen et al 2020 ICML that uses contrastive loss over augmented data, the positive and negative data pairs are required to be constructed with the synthetic and real samples.
>
> [Gidaris et al 2018 ICLR]: Unsupervised Representation Learning by Predicting Image Rotations
> [Chen et al 2020 ICML]: A Simple Framework for Contrastive Learning of Visual Representations
>
> A4 “Why the proposed method has such a good cross-architecture generalization”: While it is very difficult to provide a definite answer for the question, our intuition is that matching two sets of gradients with respect to convolutional kernels or fully connected weights enables the condensed images to encode discriminative information for the target task in a spatially structured way. At a high-level, the condensed images capture the correlation between pixels and high-dimensional deep representations in a very compressed way. We believe that such correlations are informative to the architectures that can relate these two in a similar way. For instance, our results show that condensed images learned in one convolutional architecture generalise well to another convolutional architecture but this is not the case from MLP to a convolutional network.
>
> A5 ”Data ratio for reaching 80-90% of the upper-bound in CIFAR10”: We evaluated the performance of our method on different numbers of condensed images per class in ConvNet architecture. Our method achieves 64%, 67%, 71%, 77%, 83% relative accuracy with 50, 100, 200, 500, 1000 condensed images per class, which are only 1%, 2%, 4%, 10%, 20% of the size of the original CIFAR10 dataset. We agree with the reviewer that pushing the results closer to the upper-bound is challenging and requires further improvements to our method.

---

> > ### Comment · ~Pau_de_Jorge1 · 2021-02-23
> > **Hyperparams for large synthetic sets**
> >
> > Dear authors, congratulations on this interesting work. I have looked at the code you released in https://github.com/VICO-UoE/DatasetCondensation
> >
> > And you define the hyperparams only for the dataset sizes that appear in the paper (up to 50 ipc), would it be possible to provide the hyperparams for the results reported in this comment? (with 50, 100, 200, 500, 1000 condensed images per class).
> >
> > Thank you in advance!

---

> > > ### Author Response · Authors · 2021-02-25
> > > **Hyper-parameters for extra experiments**
> > >
> > > Dear Pau, thank you for being interested in our paper! As you have opened an issue for this question in our GitHub repository, we have provided detailed hyper-parameters for these extra experiments there. Please refer to the GitHub issue.

---

### Official Review · AnonReviewer3 · 2020-10-28
**Good paper**

**Rating:** 9
**Confidence:** 4

**Review:**

The paper presents a method for generating synthetic datasets from the large realworld datasets. The CNN trained on such synthetic dataset supposed to have similar accuracy on the realworld data, as trained on the original one. The benefit of a such procedure is reduced model training time and storage space (for data).


The method is built on the idea that the gradients of the network being trained on the real images should be similar to gradients, which were obtained by the training on the synthetic images.

The method is validated on MNIST, SVHN, FashionMNIST and CIFAR-10 on several different architectures: MLP, AlexNet, VGG-like and ResNet architectures.

Moreover, the paper compares the proposed method vs  many other baselines, e.g. methods, which select "representative" image from the dataset (coreset methods), as well as Dataset Distillation and cGAN.

*****
Overall I like the paper a lot. The method is well-motivated, shows good (sota) results and also often (for MNIST, SVHN, FashionMNIST)  produces human-recognizable examples, although there is no term/regularization directly encouraging this.

The paper also studies how architectural choices like normalization, pooling, etc. influence the generated samples and how samples generated for the architecture A are suitable for training architecture B.

I don't see any major weakness in the paper.


Questions and comments.

- In Figure 5, IMO it is bad practice to fit linear regression into blob-like data. (minor)
- Have you tried to add some term, encouranding the diversity for the different synthetic samples belonging to the same class?

********

Post-rebuttal.

The rebuttal didn't raised any concerns and made the paper even stronger, thus I am keeping my score.

---

> ### Author Response · Authors · 2020-11-16
> **Response to AnonReviewer3**
>
> We thank the reviewer for his/her valuable comments and time.
>
> A1 “Fit linear regression into blob-like data”: We agree that fitting a line to such data may not be very informative, especially when the correlation is weak. Our main point was to only visualise the linear correlation between two sets of accuracies that are obtained with proxy dataset and full dataset training. We followed the previous work (Shleifer et al 2019 arXiv, Wang et al 2020 CVPR)  in the visualization.
>
> [Shleifer et al 2019 arXiv] Using Small Proxy Datasets to Accelerate Hyperparameter Search.
> [Wang et al 2020 CVPR] Nas-fcos: Fast neural architecture search for object detection.
>
> A2 “Some term for encouraging diversity”: While this is a very good suggestion, designing a good regularizer for encouraging diversity among condensed images is challenging. Naive solutions such as imposing orthogonality among the pixels of condensed images would not be meaningful. More promising solution may be encouraging orthogonality between the gradients of each synthetic sample. However, the gradient space is typically very high dimensional and a simple measure of orthogonality would be very weak. We currently work on grouping real images based on their appearance and learn a condensed image to match the average gradient of each such group.

---

> > ### Comment · AnonReviewer3 · 2020-11-18
> > **Re: A1**
> >
> > >We agree that fitting a line to such data may not be very informative...
> > >We followed the previous work...
> >
> > If previous papers proposed some poor practice, the follow-up works should correct it, not repeat and reinforce it.

---

> > > ### Author Response · Authors · 2020-11-19
> > > **Response to AnonReviewer3 - Removed the Linear Regression**
> > >
> > > Thanks for your comment. We agree with your point. Thus we have removed the linear regression from Figure F5, instead we only report the Spearman’s rank correlation coefficient. Please refer to the latest paper.

---

> > > > ### Comment · AnonReviewer3 · 2020-11-19
> > > > **Thank you for the correction!**

---

### Official Review · AnonReviewer4 · 2020-10-28
**Interesting idea with promising empirical results**

**Rating:** 8
**Confidence:** 3

**Review:**

##########################################################################

Summary:

The paper proposes a novel dataset condensation technique that generates synthetic samples by matching model gradients with those obtained on the original input dataset. This technique is investigated empirically on several smaller datasets like MNIST, SVHN and CIFAR10. Two applications to continual learning and neural architecture search (NAS) are also explored and show some promising results.

##########################################################################

Reasons for score:

Overall, I vote for accepting this paper. The technique is intuitive and well-justified. Experimental results seem to suggest that it produces a synthetic set that compares favorably to those obtained using alternative methods. Also, additional applications of this technique to continual learning and NAS appear to be quite promising.

##########################################################################

Pros:

1. The paper is well written. The core idea is arrived at systematically and is carefully explained.

2. The paper does a good job referencing prior work (with most papers that I knew of being included) and empirical results obtained by the authors appear to compare very favorably to this existing prior work. I do not think that presented empirical results are exhaustive, but they are definitely very promising.

##########################################################################

Cons.

I did not see any major problems with the paper. But wanted to make a few comments that could potentially be addressed:

1. I found a sentence "Note that each real and synthetic batch sampled from T and S contains samples from a single class and the synthetic data for each class are separately updated at each iteration" a bit confusing. Specifically, does this mean that all samples in T and S have the same label? If so, does this mean that in the case when we have only one sample per class, S contains essentially the same input sample (with or without image augmentations depending on the experiment)?

2. While S is built to match the gradients on the original architecture, it is a little counterintuitive that such a small training set would not cause dramatic overfitting on other (possibly "heavier") architectures. Do we need to use multiple synthetic sets in practice, or rely on heavy data augmentation to avoid overfitting? Or just limiting the number of training steps would be enough? It would be interesting to see a more detailed discussion of this aspect. While I am encouraged by NAS results, this remains one of my concerns.

3. It would be interesting to see results on larger datasets like ImageNet or at least CIFAR-100. I understand that this exploration would be quite computationally intensive, but this would make the results much more convincing.

##########################################################################

Questions during rebuttal period:

Please address and clarify the cons above. (I will update the score depending on the authors reply.)

#########################################################################

Post-rebuttal.

Thanks for a detailed response that clarified some questions and concerns that I previously had. I think the updated paper is stronger and I am inclined to raise the score to 8.

---

> ### Author Response · Authors · 2020-11-16
> **Response to AnonReviewer4**
>
> We thank the reviewer for his/her valuable comments and time.
>
> A1: $\mathcal{T}$ and $\mathcal{S}$ contain images from all classes. At each training iteration, we sample single-class mini-batches at Step 6 of Algorithm 1 (see the revised paper). Specifically, we sample a mini-batch pair $B^\mathcal{T}_c$ and $B^\mathcal{S}_c$ that contain real and synthetic images from the same class $c$ at each inner iteration $t$. Then, the matching loss for each class is computed with the sampled mini-batch pair and used to update corresponding synthetic images $\mathcal{S}_c$ by back-propagation (Step 7 and 8 of Algorithm 1 in the revised paper). This is repeated for every class. Training as such is not slower than using mixed-class batches, and  it can also be sped up by running them in parallel for each class. When we use mixed mini-batches, our method still works well. However, we found that separate-class strategy is faster to train as matching gradients w.r.t. data from single class is easier compared to those of multiple classes.
> We have clarified these points in our notation and comments for Algorithm 1, and also give more details in Supplementary - Implementation Details - Dataset Condensation - Paragraph 2.
>
> A2: When we use condensed images to train heavy architectures (like AlexNet, VGG, ResNet), we indeed apply substantial data augmentation (crop, scale and rotate) to avoid overfitting. This is mentioned in Supplementary - Implementation Details – Dataset condensation - Paragraph 3. We don't need to use multiple synthetic sets to train a model. We have not applied early stopping and used a fixed number of training iterations. We follow the same practice for the baselines that we compare to.
>
> A3: Thank you for understanding that we are working on a largely-unexplored and challenging research problem. Here, we give some preliminary results on the CIFAR100 dataset in the following tables. The default ConvNet is used and all hyper-parameters are the same as corresponding ones for CIFAR10. The results show that our method remarkably outperforms the strong baseline - Herding in all settings.
>
> ====================================================================
>
> Absolute accuracy % on CIFAR100 dataset:
>
> |         | 1 image/class | 10 images/class | 20 images/class |
> |----------------|---------------|----------------|----------------|
> | Random     | 4.2±0.3       | 14.6±0.5       | 20.8±0.6       |
> | Herding     | 8.4±0.3       | 17.3±0.3       |  21.5±0.3       |
> | Ours           | 12.8±0.3      | 25.2±0.3       | 30.4±0.3       |
>
> ====================================================================
>
> Relative accuracy % (the ratio compared to its upper-bound 56.2$\pm$0.3) on CIFAR100 dataset:
>
> |         | 1 image/class | 10 images/class | 20 images/class |
> |---------|---------------|----------------|----------------|
> | Random  | 7.5       | 26.0       | 37.0       |
> | Herding | 14.9       | 30.8      |  38.3       |
> | Ours    | 22.8      | 44.8       | 54.1       |

---

### Author Response · Authors · 2020-11-16
**Summary of changes in the revised paper.**

According to the reviews, we polished the paper in the following two aspects:

1. To further clarify the mini-batch sampling process, we have added several notations and comments in Algorithm 1, and also give more details and discussion in Supplementary - Implementation Details - Dataset Condensation - Paragraph 2.

2. To justify the effectiveness and robustness of the proposed gradient distance metric, we added the ablation study on the gradient distance metric in the supplementary material - Further Analysis - Ablation study on gradient distance metric.

---

### Decision · Program_Chairs · 2021-01-07
**Final Decision**

**Decision:**

Accept (Oral)

**Comment:**

The paper introduces a novel dataset condensation technique that generates synthetic samples (images) by matching model gradients with those obtained on the original input samples (images). The authors also show that these synthetic images  are not architecture dependent and can be used to train different deep neural networks. The approach is validated on several smaller datasets like MNIST, SVHN and CIFAR10. This work is well-motivated and the methodological contributions convincing. All reviewers were enthusiastic and indicated that there were no flaws in this work. The rebuttal clarified outstanding questions and made the paper stronger.